# The Ovary–Liver Axis: Molecular Science and Epidemiology

**DOI:** 10.3390/ijms26136382

**Published:** 2025-07-02

**Authors:** Ralf Weiskirchen, Amedeo Lonardo

**Affiliations:** 1Institute of Molecular Pathobiochemistry, Experimental Gene Therapy and Clinical Chemistry (IFMPEGKC), RWTH University Hospital Aachen, 52074 Aachen, Germany; 2Department of Internal Medicine, Azienda Ospedaliero—Universitaria of Modena (-2023), 41125 Modena, Italy; a.lonardo@libero.it

**Keywords:** cirrhosis, fibrosis, hypogonadism, MASLD, menopause, NAFLD, PCOS, Turner syndrome

## Abstract

In women, gonadal hormones play a crucial regulatory role in body fat distribution and glucose–lipidic homeostasis, which are closely associated with the hepatic steatogenesis and intrahepatic inflammatory pathways. Accumulating evidence supports the idea that hepatic health is closely linked to endocrine ovarian function through hormonal, metabolic, and immunological communications, collectively known as the “ovary–liver axis”. This review presents the molecular mechanisms involved in sex hormone synthesis, metabolism, and signaling pathways along the ovary–liver axis, focusing on dysregulated mechanisms that may contribute to common disorders and, specifically to hepatic derangements in the context of altered ovarian function. Additionally, we analyzed epidemiological evidence supporting the ovary–liver axis, specifically examining meta-analytic studies exploring the connection between polycystic ovary syndrome and metabolic dysfunction-associated steatotic liver disease (MASLD). We also discuss studies linking hypogonadism with liver health, with a specific focus on Turner syndrome and MASLD. Furthermore, we explore the impact of menopause on liver health. Our integrated molecular and epidemiological approach identifies important clinical and public health implications, aiming to uncover potentially innovative interventions and effective strategies for managing disease progression. However, unexplored areas within the ovary–liver axis highlight the need for further research on causal pathways.

## 1. Introduction

The concept of a functional axis connecting the ovary to the liver was first proposed in the early 1980s [1]. In recent years, the ovary–liver axis has gained significant attention as a crucial factor in women’s health. This axis involves hormonal, metabolic, and immunological interactions that connect ovarian endocrine function to liver health [2]. Estrogens and other ovarian hormones can profoundly influence glucose metabolism, lipid homeostasis, intrahepatic droplet formation, and inflammatory pathways in the liver. Meanwhile, liver function critically modulates the metabolism of these hormones, subsequently affecting ovarian processes such as folliculogenesis and ovulation [2]. Alterations in this dual interplay have been associated with conditions like polycystic ovary syndrome (PCOS), metabolic dysfunction-associated steatotic liver disease (MASLD), and metabolic syndrome (MetS), highlighting the significant burden these closely intertwined conditions can impose on overall public health [3,4,5]. While previous studies have independently explored the mutual and bi-directional influence of reproductive hormones on liver disease and the impact of hepatic dysfunction on ovarian health, a comprehensive understanding to fully explain the epidemiological patterns is still lacking.

Against this backdrop, the present review aims to first integrate current molecular insights regarding hormone synthesis, metabolism, and signaling pathways along the ovary–liver axis, with particular attention to dysregulated mechanisms that may underlie common disorders and liver disease associated with endocrine ovarian dysfunction. Second, it seeks to summarize epidemiological studies, ranging from cross-sectional surveys to population-based cohorts, and synthesize data on prevalence, risk factors, and outcomes associated with disrupted ovarian and hepatic functions. By doing so, the review will identify underexplored areas in this domain and underscore the need for further research on causal pathways. Finally, it intends to highlight the clinical and public health implications derived from combining molecular and epidemiological findings, laying out potential directions for interventions and strategies to mitigate disease progression. The present article does not cover the epidemiological and clinical underpinnings of infertility and fertility-related issues occurring in women with chronic liver disease.

For this review, a systematic literature search was conducted in major scientific databases such as PubMed, Scopus, and Web of Science. Targeted keywords included “ovary-liver axis”, “estrogen metabolism”, “reproductive hormones”, “hepatic metabolism”, “PCOS”, nonalcoholic fatty liver disease (“NAFLD”), nonalcoholic steatohepatitis (“NASH”), “MASLD”, “MASH”, metabolic dysfunction-associated steatohepatitis (“MALFD”), and “epidemiology.” Peer-reviewed articles, reviews, and meta-analyses published primarily in the past two decades were included to capture recent molecular discoveries and advanced epidemiological methods. The studies were screened for relevance based on their focus on the interplay between ovarian and liver functions. Entries without a clear link to either molecular mechanisms or epidemiological outcomes were excluded. The key data on study design, population characteristics, molecular markers, and clinical endpoints were extracted, with particular attention to the methodological robustness of epidemiological contributions. To ensure a transparent and reproducible selection process, we followed the PRISMA guidelines in screening, assessing, and reporting the findings [6] (Figure 1). Cross references and review articles were also considered when appropriate. Additional and detailed information on the strategy of bibliographic research can be found under Section 3, Section 4 and Section 5.

## 2. Molecular Mechanisms of Action of Ovarian Hormones and Role of the Ovary in Maintaining Metabolic and Liver Health

Ovarian hormones, particularly estrogens like estradiol, play a crucial role in coordinating various physiological processes that link reproductive health and systemic metabolic regulation [7]. At the molecular level, these hormones exert their effects through distinct pathways, both classical and non-classical, that collectively influence gene transcription, enzyme activity, signal transduction, and cellular metabolism. Estrogen receptors (ERs), including estrogen receptor alpha (ERα) and estrogen receptor beta (ERβ), function as nuclear transcription factors when bound by their ligands. They form receptor-hormone complexes that bind to estrogen response elements (EREs) on target genes [8]. This interaction leads to the recruitment of co-regulators such as co-activators and co-repressors, ultimately altering chromatin structure and influencing gene expression patterns. Through these genomic actions, estrogens regulate a variety of metabolic pathways, from glucose homeostasis and lipid metabolism to inflammatory signaling in multiple tissues, with the liver being one of the most responsive and significant targets [9].

In hepatocytes, ERα is particularly abundant and crucial for regulating processes that maintain metabolic health. Under normal physiological conditions, estrogen binding to ERα promotes balanced lipid metabolism by regulating the expression of lipogenic enzymes, lipoprotein receptors, and proteins involved in cholesterol homeostasis [8]. Additionally, estrogens can increase pathways responsible for lipid oxidation, decrease triglyceride accumulation, and reduce overall lipotoxic stress in liver cells [2,9]. Apart from direct gene regulation, estrogens also participate in non-classical, membrane-initiated actions by binding to G protein-coupled estrogen receptor (GPER) or by interacting with membrane-localized ERs [10]. These rapid signaling events can activate second messenger pathways, such as phosphoinositide 3-kinase (PI3K) and mitogen-activated protein kinase (MAPK), that regulate enzymatic activities, glucose uptake, and inflammatory mediators in real time, highlighting the versatility and effectiveness of estrogen signaling [11,12,13].

In addition to the general regulatory framework for lipid metabolism in hepatocytes, estrogen signaling via ERα exerts precise control over key regulatory nodes such as sterol regulatory element-binding protein 1c (SREBP-1c) and members of the peroxisome proliferator-activated receptor family (PPARs) [14,15,16]. SREBP-1 is a transcription factor that controls the synthesis of lipids from glucose in the liver [17]. Under classical signaling, estrogen-ERα complexes bind to estrogen response elements with high affinity on target genes in response to E2 [18]. Estrogen stimulates an increase in SREBP-1 expression [14], promoting de novo lipogenesis and impacting carbohydrate metabolism [19,20]. Furthermore, estrogen and PPARs exhibit complex interactions. On one side, estrogen can suppress the activity of PPARs, and, conversely, PPARs can affect estrogen-signaling pathways that promote fatty acid catabolism, suggesting a complex crosstalk exists between PPARs and ERs [21]. This balances lipid uptake, storage, and oxidation, ultimately curbing triglyceride accumulation. In concert with the non-classical pathways involving GPERs, these pathways modulate glucose uptake, insulin sensitivity, and inflammatory signaling, thereby orchestrating metabolic gene expression.

Moreover, the liver plays a crucial role in regulating estrogen metabolism (Figure 2). It conjugates and breaks down estrogens to estrogenically inactive metabolites for excretion, maintaining stable blood hormone concentrations [22]. Low estrogen concentrations and impaired estrogen signaling are associated with liver pathology in both females and males [12].

Therefore, when hepatic function is compromised, hormone clearance can be affected, leading to either low or high hormone concentrations. Both scenarios can negatively impact ovarian function and overall metabolic regulation. The ovary is responsible for synthesizing and releasing estrogen and progesterone in response to follicle-stimulating hormone (FSH) and luteinizing hormone (LH) from the anterior pituitary gland [23]. This cyclical regulation forms the basis of the menstrual cycle, with distinct phases (follicular, ovulatory, and luteal) characterized by changing hormone concentrations that prepare the body for potential pregnancy [23]. Estrogen plays a key role in folliculogenesis and ovulation as well as in maintaining systemic functions like insulin sensitivity, energy utilization, and lipid handling [24].

By controlling hormone secretion, the ovary ensures that target tissues including the liver receive optimal hormonal signals for maintaining metabolic homeostasis. The action of estrogen protects against insulin resistance (IR) and chronic inflammation, which are integral in the pathobiology of MetS [24]. Clinical observations show that—compared to postmenopausal women and men of similar age—premenopausal women have lower rates of metabolic disorders like dyslipidemia and MASLD [25]. This difference is attributed to ongoing ovarian hormone production, which specifically supports liver health in the context of systemic metabolic and immune function. Disruptions in ovarian hormone production, as seen in conditions like PCOS, premature ovarian insufficiency, or menopause, can lead to hormonal imbalances that perturb metabolic processes and impair energy and gluco-lipidic metabolism [26,27]. Elevated androgens can worsen IR in the liver and peripheral tissues [28], promoting hepatic steatosis and inflammation. Lower estrogen concentrations reduce the capacity of the liver to counteract these negative effects.

PCOS and MASLD have been correlated with infertility through overlapped pathophysiological mechanisms [29]. In women with PCOS, hyperandrogenemia can negatively impact hepatic lipid metabolism, increasing the risk of MASLD and other metabolic disorders [30]. Additionally, IR and central adiposity, common in PCOS, further strain liver function. Even in asymptomatic, otherwise healthy individuals, such as postmenopausal women, an insufficient estrogen environment can lead to weight gain, dyslipidemia, hepatic steatosis, and increased diabetes risk [31,32]. These findings emphasize the importance of maintaining ovarian function and drive research into hormone replacement therapies to protect women from hepatic and metabolic complications as they age.

In summary, the intricate relationship between ovarian hormones and metabolic/liver health underscores a complex system that regulates reproductive and metabolic balance. Estrogens play a central role in gene expression and signaling pathways, connecting energy balance, intrahepatic lipid content, and liver inflammation. The precise hormone secretion by the ovary ensures that the liver and other tissues receive the right hormonal environment, preserving metabolic health and reducing chronic disease risk. Any disruption in hormone synthesis, receptor function, or hepatic metabolism can trigger a chain reaction of disease risk, highlighting the interconnectedness of ovarian function and hepatic health. Further research into these pathways will enhance our ability to diagnose, prevent, and manage gynecological and metabolic conditions, ultimately improving health outcomes for women at all stages of life.

## 3. The Liver and PCOS

Given the abundance of published epidemiological studies, we decided to focus on meta-analytical evidence supporting an association between PCOS and NAFLD/MASLD. Our research strategy ((liver[Title/Abstract]) AND (PCOS[Title/Abstract])) AND (meta-analysis[Title/Abstract]) retrieved 17 results. Of these, nine were discarded because they focused on treatment, discussed guidelines and study protocols, were narrative review articles, or were irrelevant regarding the association of NAFLD & PCOS. The remaining eight studies are summarized in Table 1 [33,34,35,36,37,38,39,40].

Most studies have found an increased risk of NAFLD/MASLD among women with PCOS compared to PCOS-free, healthy controls [33,34,35,36,39]. These studies collectively show an association between NAFLD/MASLD and PCOS, with estimates of the strength of the association ranging from an odds ratio (OR) of 2.25 [35] to an OR of 3.93 [33]. Several mechanisms underlie this nexus, which can be both metabolic and hormonal. Studies using multivariate analysis have shown that serum androgen concentrations strongly predict NAFLD among PCOS women regardless of confounding factors [34]. Shengir et al. [36] found that IR and MetS were more common among PCOS women than among controls. Manzano-Nunez et al. [37] identified body mass index (BMI), waist circumference (WC), alanine transaminase (ALT), homeostasis model of IR (HOMA-IR), free androgen index (FAI), hyperandrogenemia (HA), and triglycerides as variables associated with significantly higher risk of NAFLD among women with PCOS. These findings have been confirmed by others [40].

While epidemiological association does not prove causality, Liu et al. [38] conducted a bidirectional two-sample Mendelian Randomization analysis using the United Kingdom Biobank genome-wide association study database. They demonstrated that genetically predicted NAFLD is causally associated with higher odds of PCOS while PCOS may not necessarily be a risk factor for NAFLD. This conclusion conflicts with clinical wisdom that generally suggests addressing the association in either direction. In 2015, the Italian Association for the Study of the Liver was the first to suggest that “*In young women, changes in the menstrual cycle and hirsutism may suggest the presence of polycystic ovary syndrome, frequently associated with NAFLD”* [41]. Conversely, evidence that NAFLD may follow a particularly aggressive course in women with PCOS [42] recommends careful assessment of liver health status in all individuals with PCOS [43]. In conclusion, additional studies are requested to ascertain whether the NAFLD-PCOS relationship is mutual and bi-directional or uni-directional such as suggested by the study published by Liu [38].

Notably, young women are typically spared from NAFLD/MASLD [44] and more advanced disease stages, including fibrosis NASH/MASH. Therefore, hyperandrogenemic PCOS impairs this typical age- and sex-related protection, making the liver more susceptible not only to fat infiltration but also to more rapid progression of inflammatory and fibrosing disease [37,45] with the inherent risks of premature mortality and dangerous hepatic and extra-hepatic outcomes [46,47]. To prevent these risks, the diagnostic strategy must be refined, also considering the lessons learned from the PCOS studies. Hyperandrogenemia plays a major role in the pathogenesis of NAFLD/MASLD in PCOS through an extended half-life of low-density lipoprotein (LDL) and very low-density lipoprotein (VLDL) via inhibited LDL-receptor expression leading to elevated concentrations of LDL, triglycerides and increased IR [39]. This pathogenic mechanism may be leveraged clinically using serum testosterone concentrations greater than 3.0 nmol/L as a cut-off value for defining PCOS women at a higher NAFLD risk [48]. At the same time, standard ultrasonography has sufficient sensitivity to detect fatty changes as low as 5% of hepatocytes [49] and, given its low cost and universal availability, remains a first-line assessment tool in this field.

Regarding the methodological limitations of the studies listed in Table 1, different diagnostic criteria have been used to identify both PCOS and NAFLD/MASLD. Liver biopsy was not utilized in these studies, and the MASLD nomenclature was not employed. Importantly, the diagnosis of PCOS relies on a mix of Rotterdam/NIH/AES criteria, which calls for additional and more standardized investigations. There is also evidence of differences among various ethnicities [36]. NAFLD exhibits a different prevalence in the general population across continents, with the highest values, around 30% for each region, being found in South America and the Middle East [27]. Gene variants like the rs738409 G allele of the patatin-like phospholipase domain-containing protein 3 gene are deemed to account for the elevated NAFLD prevalence in South America even though these individuals consume fewer daily calories than in the USA and Europe [27]. Publication bias is another concern [39]. Collectively, these drawbacks call for better designed and adequately sized additional studies.

## 4. The Liver in Hypogonadal Women

Compared to the extensive literature on PCOS, the published studies on hypogonadism and liver in women are much less numerous and there is no published meta-analytic review assessing the risk of liver disease among hypogonadic women. Therefore, a dual approach was taken to gather relevant epidemiological studies on this topic. Initially, the research strategy ((liver[Title/Abstract]) AND (hypogonadism[Title/Abstract])) AND (female[Title/Abstract]) yielded 55 results. Out of these 55 results, 7 were selected after excluding reviews, case reports, articles in languages other than English, experimental studies, articles on male hypogonadism, pharmacology of gonadotropins, radiological or pediatric aspects, and articles insufficiently focused on the liver or hypogonadism. These articles were combined with four additional articles found using the research query (ovarian insufficiency[Title/abstract]) and (Liver[Title/Abstract[) after excluding reviews, experimental articles, case reports, articles not in English, articles on menopause, retracted articles, and articles on therapeutics. Together, these eleven retrieved articles are summarized in Table 2 [50,51,52,53,54,55,56,57,58,59,60].

There seems to be a two-way relationship between chronic liver disease (CLD), particularly cirrhosis, and hypogonadism. It has long been known that cirrhosis is often accompanied by hypogonadism which, in turn, increases the risk of osteoporosis and bone fractures [50]. More recently, researchers have focused on liver health in Turner syndrome (TS). TS is a rare genetic disorder caused by a partial or complete loss of one X chromosome [61]. Its prevalence is estimated to be 1/2500 female live births [62] and its primary clinical features include short stature and premature or primary ovarian insufficiency. Other complications include heart, aortic, thyroid, and kidney diseases, neurocognitive deficiency, skeletal deformities, and impaired hearing [62].

A substantial proportion of women with TS exhibit elevated liver enzymes, most often raised gamma-glutamyl transferase (GGT) [51,54]. These altered liver tests are associated with specific karyotypic abnormalities, cardiac valve disease [51], and hormonal imbalances [59,60] in the context of a dysmetabolic systemic environment featuring the MetS and its individual components (e.g., atherogenic dyslipidemia), IR, and subclinical chronic inflammation [59,60]. A wide range of histological changes related to the NAFLD spectrum sustain these enzyme alterations, including steatosis in most cases and liver fibrosis and cirrhosis in a variable fraction of subjects regardless of obesity [54,57,58]. Non-invasive indices of fibrosis, such as fibrosis-4 (FIB-4) and aspartate transaminase to platelet ratio index (APRI), are significantly altered compared to controls [58,60]. Methodological limitations of these studies—that should be addressed in future studies—include retrospective analysis, limited sample size, and failure to assess multi-ethnic populations.

Another clinical model used to determine the association of hypogonadism with CLD involves individuals with pituitary adenomas who undergo transsphenoidal adenectomy [55]. However, this disease paradigm is complex and difficult to decipher due to multiple hormonal insufficiencies that can result in severe fibrosing CLD [63]. This may explain why a large study failed to show a statistically significant association between the prevalence of NAFLD among hypogonadal vs. eugonadal subjects with non-functioning pituitary adenoma who underwent transsphenoidal adenectomy [55]. Therefore, additional studies are needed to address this research question.

Hereditary hemochromatosis (HH) also includes features of hypogonadism and CLD [64,65]. Both conditions arise from the accumulation of excess iron in tissues, leading to end-stage functional failure of the gonads and liver unless the excess iron is removed [66]. Because both the ovaries and the liver are impacted by the primary pathogenic insult (iron accumulation and its associated oxidative stress burden), this disease model may not be entirely suitable and probably not the ideal approach for evaluating the role of hypogonadism in the development of CLD.

Alström syndrome (AS) is a monogenic form of obesity caused by recessive mutations in ALMS1 (chromosome 2q13). It also features progressive retinal dystrophy, sensory hearing loss, cardiomyopathy, type 2 diabetes, hypertriglyceridemia, and progressive hepatic and renal dysfunction in late childhood and adulthood [52]. AS is rare with an estimated prevalence of 1 to 10 in 1,000,000 people and about 700 published cases [52]. Although incompletely defined, ALMS1 is deemed to play a role in ciliary function, cell cycle regulation, endosomal trafficking, cell migration, and extracellular matrix production [52]. About half of the individuals with AS are hypogonadal, and women often have alopecia, elevated testosterone, hirsutism, oligomenorrhea and amenorrhea in adulthood [52]. As predicted by the physiology of the ovary–liver axis, subjects with AS are insulin-resistant and have higher liver enzymes and intrahepatic fat content regardless of age, sex, ethnicity, and BMI [52]. The rarity of AS makes this disease entity scarcely suitable to conduct large studies on the association of ovarian function and CLD.

## 5. The Liver in Menopause and Ovarian Senescence

Menopause, caused by the loss of ovarian follicular function and associated with a decline in estrogen concentrations, marks the end of a woman’s reproductive years. It is characterized by the absence of menstruation and typically occurs in most women between the ages of 45 and 55 as a part of normal aging [67]. In 1947, the Danish clinician Jersild, based on his observation of an outbreak of subacute liver atrophy among post-menopausal women initially attributed to viral hepatitis, was the first to suggest the notion that estrogens may play a protective effect on the liver’s health. This implies that, in post-menopausal women, the physiological decline in estrogen concentrations may make them more susceptible to severe liver disease [68]. However, much progress has been made in understanding this phenomenon since Jersild’s pioneering report. A bibliographic search using the term (LIVER[TITLE]) AND (MENOPAUSE[Title]) yielded 25 results, of which 10 were deemed relevant after excluding articles in languages other than English, reviews, and articles on therapy. Table 3 summarizes the findings of studies that assess the role of menopause in liver disease [69,70,71,72,73,74,75,76,77,78].

Although not all published studies shown in Table 3 agree, the majority strongly support the notion that menopause exposes women to more severe CLD due to the fibrosing progression of chronic hepatitis C [69] and of NAFLD/NASH and MAFLD/MASLD [70,71,73,75,77]. Additionally, subclinical coronary artery disease identified through coronary artery calcification typically occurs in women over the age of 65. This likely indicates prolonged estrogen deficiency rather than aging itself and, unlike men, is probably not strongly related to MASLD specifically [78]. Menopause typically involves body fat redistribution with a preferential accumulation of visceral adipose tissue depots which leads to an increased risk of hepatic steato-fibrotic changes due to the increased risk of MetS and its inherent alteration in glucose tolerance and lipid profile [72]. These concepts are relevant for understanding the risk of progressive NAFLD/MASLD in the context of hormone-modulating treatments such as ovariectomy [71] and tamoxifen [79].

A recent study [80] has investigated the use of Anti-Müllerian hormone (AMH) as a marker of ovarian reserve. AMH is a glycoprotein produced by ovarian follicles that gradually decreases as women age and eventually becomes undetectable at menopause. Ovarian reserve refers to the number and quality of oocytes remaining in a woman’s ovaries and is closely linked to ovarian senescence, the aging process of the ovaries [81]. In their study, Maldonado, and colleagues utilized the NASH Clinical Research Network to examine the relationship between AMH concentrations and histological outcomes of NAFLD in 205 premenopausal women, 20% of whom had PCOS. The data revealed a negative correlation between higher AMH concentrations and NASH, with these associations being more pronounced in premenopausal women without PCOS [80], suggesting that low AMH concentrations not only indicate reproductive aging but also signify worsening cardiometabolic risk profiles [82]. Further investigation is necessary to determine the clinical significance of AMH measurements in hepatological practice and research.

A comprehensive analysis of the effects of menopausal hormonal replacement therapy (MRT) on liver health is beyond the scope of this review. A recent retrospective analysis of 368 post-menopausal women exposed to MRT suggests that transdermal estrogen can be beneficial in terms of NAFLD progression [83].

## 6. Molecular Pathogenesis of Liver Disease Associated with Ovarian Senescence, Menopause, PCOS, and Female Hypogonadism

The molecular pathogenesis of liver disease-associated ovarian senescence, menopause, PCOS, and female hypogonadism is intricately connected to the intersecting endocrine and metabolic disturbances that arise from altered hepatic function and its impact on the hypothalamic–pituitary–ovarian axis [38,84]. In healthy individuals, the liver plays a pivotal role in the metabolism, conjugation, and clearance of sex hormones, most notably estrogens and androgens, as well as in the regulation of insulin signaling and systemic nutrient balance [2,9].

Elevated pro-inflammatory cytokines such as TNF-α, released by diseased liver tissue, are known to promote androgen production in ovarian granulosa cells. This disruption affects gonadotropin-releasing hormone (GnRH) pulsatility, leading to changes in LH and FSH secretion [85]. These changes contribute to an imbalance between estrogen and androgens, favoring ovulatory dysfunction, hyperandrogenism, and compromising folliculogenesis [86]. Additionally, sex hormone-binding globulin (SHBG), a glycoprotein primarily produced by the liver, plays a crucial role in binding and affecting the bioavailability of sex hormones, such as testosterone and estrogen, impacting ovarian functionality [87]. A decrease in SHBG production during hepatic insult or metabolic dysfunction results in fewer hormones being bound, leading to an increase in free active androgens, promoting hyperandrogenism and insulin resistance [87,88]. This can impair folliculogenesis, increase oxidative stress, accelerate ovarian senescence, and cause various menstrual cycle and reproductive system disturbances [87]. Furthermore, SHBG levels are significantly lower in patients with NAFLD and type 2 diabetes [89]. On the contrary, a prospective study has indicated that basal serum SHBG is negatively associated with NAFLD development but positively associated with NAFLD regression, suggesting a more complex role of SHBG in the pathogenesis of NAFLD and insulin resistance [90].

These events highlight the critical role of a dysfunctional liver in creating a cycle of hormone imbalance, inflammatory signaling, and ovarian dysfunction that contributes to conditions like PCOS, early menopause, and female hypogonadism (Figure 3).

Moreover, estrogen signaling intersects with inflammatory pathways, as elevated mediators like TNF-α and IL-6 can disrupt ER function in hepatocytes, promoting lipogenesis and contributing to steatosis [91]. Conversely, ovarian hormones, especially estrogens, help limit hepatic lipid droplet formation by suppressing lipogenic factors such as SREBP-1c and boosting fatty acid oxidation via PPARα, thus enhancing insulin sensitivity [14,15,16]. In line with this, hepatic steatosis has been reported in ERα-deficient male mice after sexual maturation [91]. Additionally, epigenetic factors such as smoking and the decreased hepatic capacity to clear estrogens after menopause further alter systemic hormone levels, potentially exacerbating inflammation, lipotoxicity, and metabolic dysregulation [92]. These mechanisms underscore the complex interplay between estrogen signaling, inflammatory mediators, and liver metabolism during the reproductive transition.

Chronic liver impairment, whether induced by viral hepatitis, alcohol-related MASLD, or autoimmune hepatopathies, can disrupt these tightly regulated processes by altering hepatic enzyme expression and activity, affecting hormone clearance, and intensifying inflammation and oxidative stress, all of which can have downstream consequences for ovarian tissue [93]. One major factor contributing to this cascade is the dysregulation of cytochrome P450 enzymes in the liver, which typically break down estrogens and other steroid hormones into inactive metabolites for elimination [22]. When these enzymes are compromised, either through reduced expression or functionality, circulating hormone concentrations can deviate significantly from normal ranges. Excess estrogen can trigger an abnormal feedback loop at the hypothalamus and pituitary level, leading to changes in gonadotropin release, while inadequate estrogen, or an elevated androgen-to-estrogen ratio, can accelerate ovarian aging and anovulatory cycles seen in hypogonadal women or with premature menopause. Moreover, abnormal expression of estrogen and its receptors has been associated not only with ovarian disorders but also with malignant tumors [93]. Concurrently, inflammatory cytokines such as TNF-α, IL-6, and IL-1β, often elevated in liver disease, can enter systemic circulation and target various tissues, including the ovaries. There, they can trigger stress responses that impede follicle development, disrupt corpus luteum formation, and promote apoptotic pathways in granulosa cells, potentially driving ovarian cancer development [94]. Over time, these inflammatory injuries contribute to the gradual decline of ovarian follicles, reflected in clinical manifestations of early menopause or reduced ovarian reserve in younger women.

An intricate relationship between hepatic insulin regulation and ovarian function also plays a role in the molecular pathogenesis of liver disease-related ovarian conditions such as PCOS [95]. IR, a characteristic of many chronic liver diseases, can either initiate or worsen hepatic steatosis, leading to hyperinsulinemia [96]. This can boost androgen production in the gonads and suppress hepatic production of SHBG [97]. Reduced SHBG allows free testosterone to accumulate, shifting the hormonal balance towards hyperandrogenism, a key feature of PCOS [98]. This hyperandrogenic environment disrupts follicular growth and ovulation in the ovaries, while also triggering a persistent low-grade systemic subclinical inflammatory state. Elevated free fatty acids, originating from insulin-resistant adipose tissue and the diseased liver, further perpetuate this inflammatory environment by releasing pro-inflammatory substances and activating stress signals in ovarian theca cells [99]. Additionally, high insulin concentrations and hepatic dysfunction may converge on ovarian tissues through the mammalian target of rapamycin (mTOR) pathways [100]. It is worth noting that mTOR inhibitors can enhance cell growth and potentially contribute to ovarian cyst development [101]. In such conditions, excess reactive oxygen species (ROS) produced during liver and ovarian inflammation can harm mitochondrial structure, DNA, and function, thereby exacerbating the aging and apoptotic decay of oocytes [102]. Consequently, while PCOS is typically characterized by hyperandrogenism and chronic anovulation, prolonged disease duration and compounded stressors can lead to a decline in overall ovarian reserve, pushing women towards earlier ovarian failure or a more severe hypogonadal state [43].

Menopause, either premature or physiological, is influenced by these pathophysiological overlaps. Pre-menopause and menopause show a gradual decline in ovarian estrogen production to concentrations equal to those of age-matched men, which removes a protective buffer against metabolic, oxidative, and inflammatory insults in the liver and other tissues [103]. While menopause can occur independently of liver disease as part of natural aging, chronic hepatitis or cirrhosis can accelerate follicular atresia by increasing catabolic and inflammatory pathways that damage ovarian tissue. In particular, there is a unique hepato-ovarian link in which the liver and the ovaries develop similar metabolic abnormalities during the pathogenesis of MASLD and PCOS [104]. In menopause, the hepatic metabolism of estrogens, already reduced by the declining hormonal production of the ovaries, may be further compromised by cirrhotic changes that impede blood flow and impair enzyme function. Therefore, menopausal women with underlying liver disease may face a heightened risk of metabolic comorbidities, bone density loss, and cardiovascular complications [105]. Additionally, the combination of low estrogen concentrations, systemic inflammation, and potential hyperandrogenism can exacerbate IR, creating a vicious cycle where the decline in hepatic and ovarian health mutually reinforces each other. This interdependence is particularly evident in individuals with genetic predispositions to autoimmune disorders or lifestyles that increase metabolic risks, such as excessive alcohol consumption, cigarette smoking, obesity, or prolonged exposure to environmental toxins, all of which can intensify the cascade leading to liver disease [106,107]. This can potentially lead to gonadal senescence and metabolic collapse due to the aforementioned ovary–liver connection.

Female hypogonadism, whether primary (due to intrinsic ovarian pathology) or secondary (stemming from pituitary or hypothalamic dysfunction), is another manifestation that can emerge from or be exacerbated by liver disease processes [106]. Reduced hepatic clearance of toxins and byproducts of metabolism may lead to an altered endocrine environment around the hypothalamus, interfering with the pulsatile secretion of GnRH that is required for the cyclical release of LH and FSH [108]. Over time, this dysregulated gonadotropin profile impedes the ovarian production of sex steroids, potentially leading to a hypogonadal scenario. Additionally, hepatic complications can impede the normal synthesis of steroid precursor proteins and transport molecules, such as albumin and hormone-binding globulins, leading to circulating imbalances in estradiol, progesterone, and androgens [109]. The resulting hormonal disequilibrium impairs ovulatory cycles and fosters detrimental feedback loops that can lock the ovaries into a state of persistent dysfunction and accelerate follicular depletion. Furthermore, chronic alterations in amino acid metabolism and micronutrient absorption, both of which are frequently encountered in advanced liver disease, may limit the availability of essential substrates for steroidogenesis, compounding the hypogonadal state. Consequently, high macronutrient intakes with low micronutrient content are associated with the development of overweight and obesity [110], again impacting the ovary–liver axis.

Ultimately, ovarian senescence, menopause, PCOS, and female hypogonadism represent partially overlapping endpoints on a continuum of disrupted inter-organ communication. Hepatic insufficiency and inflammation derail the finely tuned hormonal networks that govern reproductive function. From a molecular standpoint, the chronic supply of pro-inflammatory cytokines triggering dysregulated steroidogenic pathways, and the buildup of oxidative and metabolic stress in both the liver and the ovaries play a central role in this deterioration [93]. As negative feedback loops multiply over time, the depletion of functional follicles occurs, pushing women towards premature ovarian senescence. Recent whole-organ imaging conducted in middle-aged mice has demonstrated that vascular aging in the ovaries is a significant factor associated with ovarian aging and decreased fertility [111].

Understanding these molecular underpinnings not only sheds light on the etiology of gynecological disorders in the context of liver disease but also provides a rationale for targeted interventions to intercept or slow the path towards endocrine failure [112]. These interventions may include anti-inflammatory agents, insulin sensitizers, antioxidants, carefully balanced hormone replacement therapies to prevent liver damage, or novel strategies aimed at restoring the normal hepatic metabolism of sex hormones. Furthermore, ongoing research into biomarkers such as altered concentrations of bone turnover biomarkers, hormone concentrations, growth hormone, markers of ovarian reserve such as AMH, and many others that capture the earliest shifts in liver–ovary crosstalk holds the promise of enabling clinicians to identify at-risk women well before profound ovarian senescence or hypogonadism develop, allowing for preventive or mitigative measures [106]. This work has recently shown that there are several shared markers between NAFLD and PCOS related to circadian rhythm disruption [113].

There are several potential strategies that could be highly beneficial in disrupting the mutual connection between NAFLD and PCOS. Initial therapeutic interventions such as lifestyle modifications including diet, weight loss and exercise are considered the simplest ones [114]. However, there are also different pharmacologic strategies, include antioxidant therapies aimed at reducing oxidative stress and hepatocellular damage, insulin sensitizers (e.g., metformin, thiazolidinediones, glucagon-like peptide-1 receptor agonists) that restore metabolic homeostasis and improve gonadal hormone balance, and mTOR inhibitors such as miR-615-5p that could attenuate hyperproliferative signals, inflammatory cascades, and decrease lipid droplet accumulation [30,115,116,117]. Exploring these strategies will enhance our understanding, highlighting not only their potential in reducing liver injury but also their relevance in addressing ovarian dysfunction and mitigating reproductive decline.

## 7. Conclusions

A mutual and bi-directional interaction occurs between liver health and ovarian function. While the consequences of underlying CLD on female fertility have been examined elsewhere [118], here, we have adopted a combined analysis of molecular science and epidemiological evidence to specifically identify the scope of the ovary–liver axis across a spectrum of disorders and conditions. These range from physiological and premature menopause to PCOS and menopause. By understanding the molecular intricacies of how a diseased liver can undermine ovarian function, and conversely, how disrupted endocrine ovarian function may eventually impair liver health, clinicians and researchers can develop more comprehensive approaches to women’s health. These approaches ensure that interventions are holistically designed to maintain not only reproductive capability but, importantly, also metabolic health, duration and quality of life. This knowledge not only informs clinical science but also shapes public health interventions and policies.

## Figures and Tables

**Figure 1 ijms-26-06382-f001:**
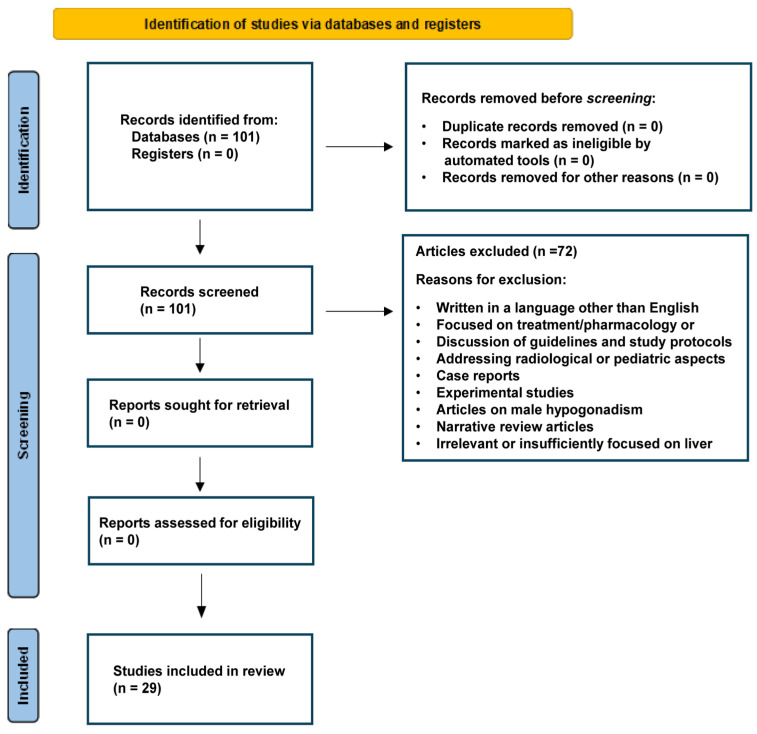
PRISMA flow diagram depicting the study screening and selection process. The diagram was prepared following proposed recommendations [6].

**Figure 2 ijms-26-06382-f002:**
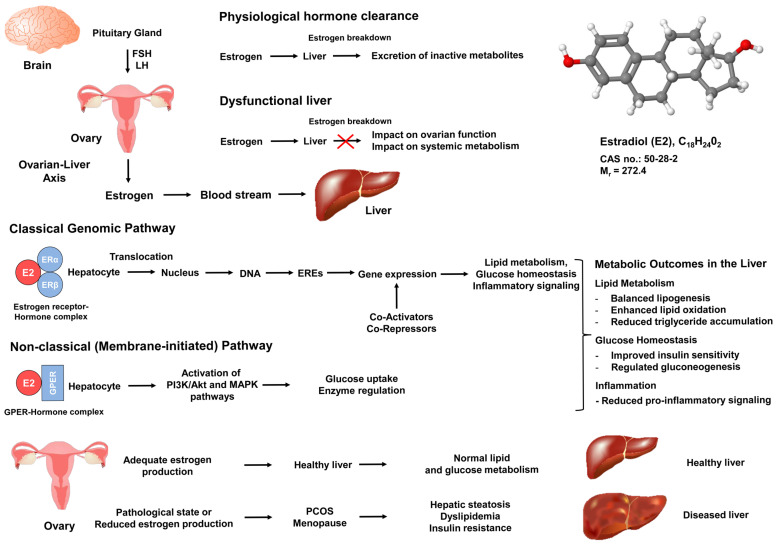
Estrogen-signaling pathways and the ovary–liver axis in metabolic regulation. Estrogen signals originating from the ovary impact metabolic and inflammatory pathways in the liver through classic (genomic) and non-classic (membrane-initiated) mechanisms. Gonadotropins (FSH and LH) regulate estrogen release from the ovary, which then circulates to hepatocytes. Estradiol (E2), the most biologically active form of estrogen, binds to estrogen receptor alpha (ERα) or estrogen receptor beta (ERβ), forming hormone-receptor complexes that translocate to the nucleus and bind estrogen response elements (EREs) to regulate gene transcription. Additionally, estradiol can bind to the G protein-coupled estrogen receptor (GPER) on cell membranes, activating rapid signaling cascades (e.g., PI3K, MAPK) that control processes like glucose uptake, lipid oxidation, and inflammation. Key metabolic outcomes are depicted, such as improved lipid metabolism, reduced triglyceride accumulation, and decreased pro-inflammatory cytokine signaling, demonstrating how estrogens support healthy liver function and overall systemic balance.

**Figure 3 ijms-26-06382-f003:**
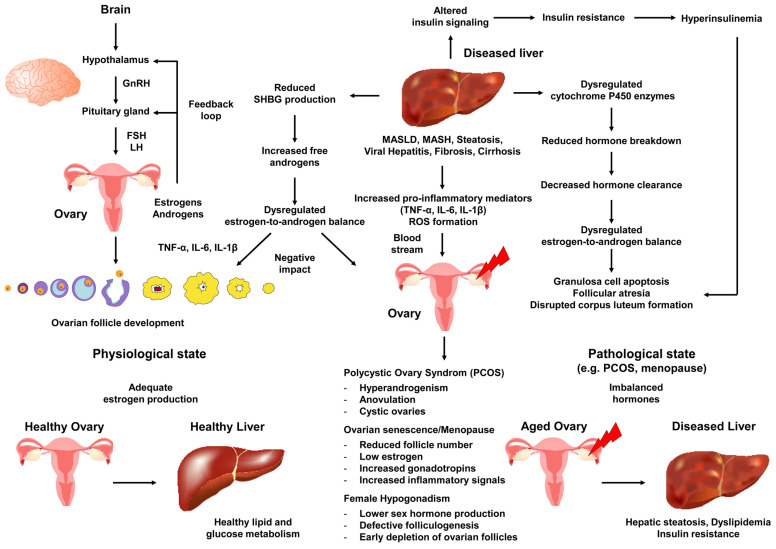
Hepato-ovarian crosstalk in the molecular pathogenesis of female reproductive disorders. The diagram displays the bidirectional influences between the liver and ovaries in the context of chronic liver disease. A liver affected by conditions such as MASLD, cirrhosis, or viral hepatitis undergoes changes in cytochrome P450 enzyme activity, reduced hormone-metabolizing capacity, increased inflammatory cytokines, and insulin resistance. These changes impact the concentrations of circulating estrogens and androgens, while also favoring subclinical systemic inflammation. The hypothalamic–pituitary–ovarian (HPO) axis transmits signals through gonadotropin-releasing hormone (GnRH), follicle-stimulating hormone (FSH) and luteinizing hormone (LH) receiving distorted feedback signals due to abnormal hormone ratios (e.g., higher androgen-to-estrogen concentrations) and inflammatory mediators circulating through the bloodstream to various organs. The ovary transitions with aging into pathophysiological states such as PCOS, early menopause, or hypogonadism (indicated by a red arrow). Key disruptions caused by the diseased liver affecting the ovary include hyperinsulinemia driving hyperandrogenism, reduced sex hormone-binding globulin (SHBG) production, and compromised folliculogenesis, all contributing to increased oxidative stress and follicular atresia. The bottom panels emphasize how sustained hepatic insults accelerate ovarian senescence, highlighting how impaired clearance by the liver and metabolic dysregulation of hormones can worsen ovarian dysfunction, creating a vicious cycle that may contribute to the development of PCOS, reproductive decline, or hypogonadism. Abbreviations are as follows: IL-1β, interleukin-1β; IL-6, interleukin-6; TNF-α, tumor necrosis factor-α.

**Table 1 ijms-26-06382-t001:** Meta-analytical evidence supporting associations between PCOS and NAFLD/MASLD.

Author(s), Year [Ref]	Method	Findings	Conclusion
Ramezani-Binabaj et al., 2014 [33]	In total, 7 eligible studies were included, totaling 616 PCOS cases (defined according to the Rotterdam criteria) and 569 healthy controls. NAFLD was assessed with USG.	NAFLD is more prevalent among PCOS subjects than in healthy controls (OR 3.93, 95% CI: 2.17, 7.11).	Women with PCOS have an increased risk of NAFLD.
Rocha et al., 2017 [34]	In total, 17 eligible studies were included, totaling 2734 PCOS individuals (diagnosed based on Rotterdam criteria except for two studies using androgen excess and PCOD society, one study using the NIH criteria, and one study with missing information was NA) and 2561 controls of similar age and BMI. The criteria for diagnosing NAFLD were NA.	PCOS subjects exhibit an increased prevalence of NAFLD (OR 2.54, 95%, CI 2.19–2.95).	Women with PCOS have an increased prevalence of NAFLD. This is linked to high serum total testosterone, FAI, obesity, and IR.
Wu et al., 2018 [35]	In total, 17 eligible studies were included, totaling 2715 PCOS subjects (diagnosed based on Rotterdam criteria except for two studies using androgen excess and PCOD society, one study using the NIH criteria, and one study with missing information was NA) and 2619 controls. The criteria for diagnosing NAFLD were NA.	PCOS individuals have a higher prevalence of NAFLD than controls (OR 2.25, 95%, CI = 1.95–2.60). HA is a risk factor for NALD among those with PCOS (OR 3.31, 95% CI 2.58–4.24).	PCOS is associated with a higher risk of NAFLD, likely due to HA.
Shengir et al., 2021 [36]	In total, 23 eligible studies were included, totaling 4162 PCOS women and 2983 controls. The diagnosis of PCOS was based on one of the following definitions: Rotterdam criteria, NIH criteria, or AES criteria. NAFLD was diagnosed with imaging studies or noninvasive biomarkers.	PCOS women have a higher risk of NAFLD than controls (OR 2.49, 95%, CI 2.20–2.82). South American/Middle East PCOS patients exhibited a greater risk of NAFLD (OR 3.55, 95% CI 2.27–5.55) compared to their European and Asian counterparts (OR 2.22, 95% CI 1.85–2.67 and OR 2.63, 95% CI 2.20–3.15, respectively). IR (OR 1.97, 95% CI 1.44–2.71 and MetS (OR 3.39, 95% CI 2.42–4.76) were more frequent among PCOS women than among controls.	Premenopausal women with PCOS have a 2.5-fold increased risk of NAFLD, with BMI playing a significant role in this association.
Manzano-Nunez et al., 2023 [37]	In total, 36 eligible studies were included, totaling 5021 individuals with PCOS (diagnosed by NIH, Rotterdam, or AES criteria) and 2156 controls. NAFLD was identified either histologically or non-invasively.	MA of proportions found a pooled NAFLD prevalence of 43%, 95% CI 35–52%, with high heterogeneity. BMI, WC, ALT, HOMA-IR, FAI, HA, and TG were associated with significantly higher risk of NAFLD among women with PCOS.	NAFLD is common in young women with PCOS, influenced by both metabolic and PCOS-specific endocrine factors.
Liu et al., 2023 [38]	Bidirectional two-sample MR analysis was conducted using glycemic-related traits in up to 200,622 individuals and sex hormones in 189,473 women from a large-scale biopsy-confirmed NAFLD UKB GWAS database.	Subjects with a higher genetic propensity to NAFLD were at increased risk to develop PCOS (OR per one-unit log odds increase in NAFLD: 1.10, 95% CI: 1.02–1.18; *p* = 0.013). MR mediation analysis showed indirect causal effects of NAFLD on PCOS via fasting insulin alone (OR 1.02, 95% CI 1.01–1.03; *p* = 0.004) and in concert with androgen concentrations.	While genetically predicted NAFLD is linked to a higher likelihood of PCOS, the idea that PCOS may lead to NAFLD is not well-documented, suggesting the need for additional studies.
Yao et al., 2023 [39]	In total, 32 studies were included, totaling 145,131 PCOS patients and 50,832,503 controls. NAFLD was assessed with USG	PCOS was associated with a high risk of NAFLD (OR 2.93, *p* < 0.001, 95% CI 2.38–3.62). Age and BMI did not explain heterogeneity across the studies while a significant risk of publication bias was found.	PCOS is associated with a high risk of NAFLD.
Wu et al., 2025 [40]	In total, 26 eligible studies (all from China except one from Korea) were included, totaling 4510 participants. The diagnosis of PCOS was based on Rotterdam criteria in all studies except for five using the PCOS-PRCHIS criteria. MASLD was identified with USG in all studies except one where the criterion was NA.	At the pooled MVA, WHR (*p* < 0.001), testosterone (*p* = 0.034), and HOMA-IR (*p* = 0.02) were significantly greater in PCOS women with MASLD.	In East Asia, MASLD is associated with obesity, IR, and HA in women with PCOS.

Abbreviations used: AES—androgen excess and PCOS society; ALT—alanine transaminase; BMI—body mass index; CI—confidence interval; FAI—free androgen index; GWAS—genome-wide association study; HA—hyperandrogenism; HOMA—homeostasis model assessment; IR—insulin resistance; MA -meta-analysis; MASLD—metabolic dysfunction-associated steatotic liver disease; MVA—multivariate analysis; MR—Mendelian randomization; MetS—metabolic syndrome; NAFLD—nonalcoholic fatty liver disease; NA—not addressed; NIH, national institutes of health; OR—odds ratio; PCOS—polycystic ovary syndrome; PRCHIS—People’s Republic of China health industry standard; TG—triglycerides; UKB—United Kingdom Biobank; USG—ultrasonography; WC—waist circumference; WHR—waist-to-hip ratio.

**Table 2 ijms-26-06382-t002:** Epidemiological evidence supporting the association between female hypogonadism and liver disease.

Author(s), Year [Ref]	Method	Findings	Conclusion
Diamond et al., 1990 [50]	A comparative study was conducted on 115 individuals with biopsy-proven CLD and 113 age- and sex-matched controls.	LC and hypogonadism were risk factors for both spinal (β coef = 0.190 and 0.176; SE = 0.079 and 0.086, respectively) and forearm osteoporosis (β coef = 0.20 and 0.29; SE = 0.073 and 0.80, respectively). SBD was the predominant determinant of spinal fractures (β coef = −0.007; SE = 0.001), while hypogonadism (β coef = 0.363; SE = 0.075) and LC (β coef = 0.185; SE = 0.068) predicted peripheral fractures.	Hypogonadism is strongly linked to osteoporosis and peripheral fractures in individuals with biopsy-proven CLD.
Calanchini et al., 2018 [51]	A retrospective assessment was performed on 125 TS women.	RLE (most often altered GGT values) were identified in 49.6% of cases, and individuals exhibiting RLE had greater diameters of the sinuses and ascending aorta. A higher prevalence of RLE was found among women with isochromosome of the X long arm whereas patients with 45,X/46,XX, 45,X/47,XXX or 45,X/46,XX/47,XXX exhibited a lower prevalence of abnormal GGT values. FIB-4 > 1.3 was found in 11.8%. Fibroscan suggested significant LF in 38.1% of cases and histological changes (including 2 with LC) were found in 45.4% of 11 liver biopsies.	RLE are frequently seen in TS women, especially in connection with specific karyotype abnormalities and aortic dilatation, and may indicate fibrosing liver disease.
Han et al., 2018 [52]	A total of 38 patients with AS (ranging in age from 2 to 38 years) were compared to 76 controls matched for age, sex, race, and BMI.	AS individuals exhibited significantly higher fasting and MMT IR indices, higher MMT glucose, insulin, and C-peptide values, higher HbA1c, and TG, higher prevalence of T2DM (*p* < 0.001) lower HDL-Chol, and a 10-fold higher prevalence of MetS (*p* < 0.001), a significantly greater steatosis extent and higher transaminase values (*p* < 0.001).	Severe IR, T2DM, and SLD are the hallmarks of AS.
Viuff et al., 2021 [53]	A total of 1156 individuals with TS identified with the DCCR were compared to 577 age-matched female controls.	Women with TS have a 13-fold (IRR 12.9 (95% CI 5.8–28.8)) increased risk of liver disease owing to TLD (IRR 8.0 (95% CI 1.8–35.4)), liver insufficiency (IRR 6.7 (95% CI 1.7–26.9)), LF/LC (IRR 16.5 (95% CI 2.2–122.1)) and unspecified liver disease (IRR 10.6 (95% CI 4.4–25.3)). Furthermore, the presence of RLE was increased 12-fold (IRR 12.4 (95% CI 4.2–36.6)).	TS individuals have a high risk of liver disease, with a potential beneficial effect of HRT on liver diseases.
Bourcigaux et al., 2023 [54]	A single-center, retrospective cross-sectional analysis of 264 TS individuals.	In total, 42.8% of these individuals had raised liver enzymes. However, fewer than 10% were at risk of developing LF and LC was observed in 2 out of 19 liver biopsies.	Approximately 10% of TS subjects are at risk of LF, and the FIB-4 score should be included in the screening strategy.
Hwang, et al., 2023 [55]	A total of 278 individuals with NFPA who underwent transsphenoidal adenectomy were enrolled.	Gonadal function was unassociated with the prevalence of NAFLD among NFPA subjects (29.3% eugonadic vs. 47.8% hypogonadic, *p* = 0.14).	Hypogonadism is not linked to NAFLD prevalence in NFPA subjects.
Lam et al., 2023 [56]	In total, 68 TS patients were recruited.	Liver disease was found in 4.4% of cases, more often among individuals with Y structural rearrangement than among those with complete X monosomy (8.3% vs. 4.2% *p* = 0.771).	Steatohepatitis and inflammatory liver disease can be present in TS individuals.
Twohig et al., 2023 [57]	In total, 55 TS and 50 controls were enrolled.	Compared to controls, women with TS more often had steatosis (65% vs. 12%, stage 1 vs. 0, *p* < 0.0001) and fibrosis (39% vs. 2%, average Metavir F2 vs. F0, *p* < 0.00001) irrespective of BMI) (*p* < 0.01). GGT is more sensitive than AST or ALT in identifying liver changes.	Compared to healthy controls, TS is associated with a higher risk of SLD which can be identified by GGT serum values and SWE.
Zaegel et al., 2024 [58]	In total, 66 patients with TS were compared to 66 healthy controls using matched-pair analyses.	At LRA, TS was significantly associated with ALRI, APRI, GPR and liver dysfunction.	ALRI, APRI, and GPR show promise as biomarkers of liver disease in women with TS.
Robeva et al., 2024 [59]	A retrospective analysis of 28 healthy women, 77 individuals with DOR; and 121 patients with POI (of whom 36 had TS and 85 had non-TS POI).	Compared to controls, women with DOR, non-TS POI, and TS had RLE, pronounced IR, and worse lipidemic asset (*p* < 0.008 for all). Moreover, TS subjects had significantly higher AAT, GGT, and TSH concentrations compared to non-TS POI and DOR individuals.	Regardless of the severity of endocrinological issues and karyotypic abnormalities, decreased ovarian function is connected to IR, dyslipidemia, and RLE.
Ridder, et al., 2025 [60]	In total, 82 women with TS and 59 female controls were studied.	TS subjects showed higher values of GGT, AST, ALT (which were correlated with the inflammatory biomarkers CRP and sCD163 and 11β-hydroxytestosterone concentrations) and FIB-4 than controls (*p* < 0.001, all). Additionally, the neutrophil activation marker MPO was elevated in TS and correlated with liver parameters and sCD163.	In TS, RLE are associated with low-grade chronic inflammation and hormonal imbalances.

Abbreviations used: ALP—alkaline phosphatase; ALRI—aspartate transaminase to lymphocyte ratio index; ALT—alanine transaminase; APRI—aspartate transaminase to platelet ratio index; AS AlstrÖm syndrome; AST—aspartate transaminase; raised liver enzymes—ALP; ALT, AST, or GGT > 1.5 the ULN; Body mass index—BMI; CLD—chronic liver disease; CRP—C-reactive protein; DCCR—Danish Cytogenetic Central Registry; DOR—diminished ovarian reserve; FIB-4—fibrosis 4 index; GGT—gamma-glutamyl transferase; GPR—GGT to platelet ratio; HRT—hormone replacement therapy; IR—insulin resistance; IRR—Incidence Rate Ratio; LF—liver fibrosis; LC—liver cirrhosis; LRA—logistic regression analysis; MMT mixed meal test; MPO—myeloperoxidase; NFPA nonfunctioning pituitary adenomas; POI—premature ovarian insufficiency; RLE—raised liver enzymes; SBD Spinal bone density; sCD163—soluble CD163; SLD—steatotic liver disease; SWE—shear-wave elastography; TLD—toxic liver disease; T2DM—type 2 diabetes; TS—Turner syndrome; ULN—upper limit of normal.

**Table 3 ijms-26-06382-t003:** Epidemiological studies assessing the risk of chronic liver disease in menopause.

Author(s), Year [PMID]	Method	Findings	Conclusion
Codes et al., 2007 [69]	In total, 251 women were enrolled, of which were 22 menopausal and 65 received HRT.	Women with F2-F4 Metavir scores had a known infection duration of >15 years, higher BMI and more frequent steatosis compared to thosie with F0-F1 LF. They were more likely to be menopausal and less likely to receive HRT. Steatosis was more common and severe in menopausal women.	Factors associated with the severity of LF include a duration of infection of more than 15 years, a higher BMI, severe steatosis, and menopause. Additionally, HRT in menopausal woman is linked to a lower LF stage in CHC.
Yang et al., 2014 [70]	In total, 541 adults with biopsy-proven NASH were enrolled in the study.	After multiple adjustments (ACOR) and 95% CI, the risk for more severe LF was 1.4 (0.9, 2.1) for postmenopausal women (*p* = 0.17) and 1.6 (1.0, 2.5) for men (*p* = 0.03, compared to premenopausal women. The risk of greater fibrosis severity in men compared to women was 1.8 (1.1, 2.9) for patients <50 years (*p* = 0.02) and 1.2 (0.7, 2.1) for those ≥50 years (*p* = 0.59).	Compared to pre-menopausal women, men are at a higher risk of developing more severe LF, while postmenopausal women have a similar risk of LF severity compared to men.
Matsuo et al., 2016 [71]	A retrospective study was conducted on 666 endometrial cancer cases that underwent surgical staging, along with 209 endometrial hyperplasia cases that underwent hysterectomy-based treatment. This study also included 712 oophorectomy cases and 163 nonoophorectomy cases.	Oophorectomy was strongly associated with NAFLD risk (HR, 1.70; 95% CI, 1.01–2.86; *p* = 0.047) and NAFLD was significantly associated with postoperative T2DM (HR, 2.26; 95% CI, 1.52–3.35; *p* < 0.0001) and hypercholesterolemia (HR, 1.71; 95% CI, 1.12–2.63; *p* = 0.014).	In young women with endometrial cancer, oophorectomy is significantly associated with an increased NAFLD risk due to post-operative T2DM and hypercholesterolemia.
Veronese et al., 2018 [72]	An analysis was performed on 752 women in menopause and 535 in pre-menopause.	The years from menopause were not associated with the severity of NAFLD (*p* for trend = 0.74; Spearman correlation = 0.04; 95% CI: −0.09 to 0.17), whereas all the indexes of adiposity and the number of MetS components were associated with a higher liver steatosis score.	The higher prevalence of NAFLD observed post-menopausally may be attributed to abdominal adiposity and MetS, rather than menopause itself.
Park et al., 2020 [73]	A study was conducted on 4354 postmenopausal women who participated in the 2010–2012 KNHANES.	The OR for NAFLD per 1-year increase in age at menopause was 1.01 (95% CI, 0.99–1.03; *p* = 0.329). The prevalence of advanced fibrosis was 2.1% (95% CI, 0.7–6.4%), 2.2% (95% CI, 1.3–3.8%), and 3.9% (95% CI, 1.2–12.2%) in early (<45 years), normal (45–54 years), and late (≥55 years) menopausal women, respectively.	While there is no evidence linking early menopause to the risk of NAFLD, this study demonstrates that advanced fibrosis related to NAFLD is highly prevalent post-menopause.
Jaroenlapnopparat et al., 2023 [74]	A metanalysis was performed of 12 published studies.	Menopause and NAFLD are significantly associated (pooled OR 2.37, 95% CI, 1.99–2.82) and this association remained significant in a sensitivity meta-analysis of six studies with adjustment for age and metabolic factors (pooled OR 2.19, 95% CI, 1.73–2.78), without any evidence of publication bias.	Menopause doubles the risk of NAFLD.
Raverdy et al., 2023 [75]	In total, 1446 participants with obesity were enrolled in this study.	NASH and F ≥ 2 prevalence was 15.4% (33/215) and 15.5% (32/206) among premenopausal women with T2DM vs. 29.5% (33/112) and 30.3% (N = 36/119) in postmenopausal women with T2DM (*p* < 0.01). The distinct contribution of menopause was proven by the interaction between sex and age with respect to NASH among T2DM patients (*p* = 0.048).	A notably high prevalence of advanced SLD occurs after menopause among women with T2DM.
Kim et al., 2024 [76]	In total, 1888 participants were enrolled and followed for a median of 12.3 years.	After adjusting for confounding factors, the HR for new-onset MAFLD was 1.40 (CI 1.00–1.95) in women with menopause at <40 years compared to those in whom menopause occurred at the age of ≥50 years.	The risk of MAFLD risk is higher in women with premature menopause (under 40 years) compared to those who experienced menopause at 50 years or older.
Yang et al., 2025 [77]	In total, 1316 postmenopausal and 3049 premenopausal women were enrolled between 2006 and 2017 and followed up till 2021.	Compared to premenopausal subjects, women who experienced menopause exhibited higher chances of NAFLD (9-year aHR = 1.219, 95%, CI 1.088–1.365). BWL of ≥3% or WC reduction by ≥5% was associated with a 31.1% reduction (95% CI, 20.8–40.0%) or a 14.2% reduction (95% CI, 1.1–25.6%) in the risk of NAFLD among premenopausal women.	Menopause is associated with an increased risk of NAFLD, partly due to the accumulation of visceral fat.
Bagheri et al., 2025 [78]	In total, 446 patients undergoing CT coronary scanning were included in this study.	MASLD and CKD increased CAC risk in male but not female patients, with menopause significantly modifying LKMH’s effect.	The impact of LKMH on CAC burden is significantly influenced by liver fat content and menopause,

Abbreviations used: ACOR—adjusted cumulative odd ratio; aHR—adjusted hazard ratio; BMI—body mass index; BWL—body weight loss; CAC—coronary artery calcification; CI—confidence interval; CHC—chronic hepatitis C; CT—computed tomography; HRT—hormonal replacement therapy; KNHANES—Korea National Health and Nutrition Examination Survey; LKMH—liver-kidney metabolic health; LF—liver fibrosis; MAFLD—metabolic dysfunction-associated fatty liver disease; MASLD—metabolic dysfunction-associated steatotic liver disease; MetS—metabolic syndrome; NAFLD—nonalcoholic fatty liver disease; NASH—nonalcoholic steatohepatitis; OR—odds ratio; T2DM—type 2 diabetes; WC—waist circumference.

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
