# Peer review of "The Ovary–Liver Axis: Molecular Science and Epidemiology"

_ijms, 2025, doi:10.3390/ijms26136382_

Round 1

Reviewer 1 Report

Comments and Suggestions for Authors 1. Language & Logic (AI Editing Caveat): Although the manuscript's logic and language are generally accurate (the authors note the text was AI-polished), numerous subtle errors persist throughout. These must be corrected prior to publication. The authors are strongly advised to meticulously proofread the entire manuscript, as the accuracy of AI-generated text cannot be guaranteed. 2.High Similarity Index: The manuscript exhibits a high similarity index. Even though this is a systematic review of prior research, the similarity rate should be controlled below 25%. 3.Diagnostic Criteria Limitations: The diagnosis of PCOS relies on a mixed Rotterdam/NIH/AES criteria (Table 1), and NAFLD/MASLD is diagnosed via ultrasound (not biopsy). The potential impact of these diagnostic methods on the results (e.g., false-negative rates) needs to be explicitly discussed. 4.Addressing Causal Inconsistency: The Mendelian randomization study by Liu et al. (2023) suggests a unidirectional causal effect of NAFLD → PCOS. However, the apparent contradiction with conventional understanding (Section 7) is not adequately explained. It is recommended that the authors supplement this discussion by considering potential confounding factors, such as the mediating role of insulin resistance. 5.Undefined Abbreviations: Abbreviations such as FIB-4, APRI, and FAI are used in the text but are not defined in the list of abbreviations. 6.Mechanistic Detail Required (Fig 3): The discussion on how hepatic metabolic disturbances (e.g., elevated inflammatory factors TNF-α/IL-6) affect ovarian function via the hypothalamic-pituitary axis needs greater mechanistic detail. Figure 3 should be annotated to highlight key molecules involved in this pathway. 7.Specific Pathways Needed: The description of how classical/non-classical pathways (e.g., ERα/GPER) regulate hepatocyte lipid metabolism should be more specific. Instead of a general overview, detail the regulation of specific pathways (e.g., SREBP-1c, PPARα pathways). 8.Figure Quality (Fig 3): The resolution/clarity of Figure 3 is insufficient and must be improved. 9.Figure Quality (Fig 2): Figure 2 suffers from the same clarity/resolution issues as Figure 3. 10.Critical AI-Generated Content Warning (Reiteration): It is imperative to reiterate that significant portions of this manuscript appear AI-generated. Accuracy must be rigorously ensured; failure to do so risks propagating misinformation that could mislead subsequent research.

Author Response

Dear Reviewer 1,

Thank you for your valuable comments and the time you spent reviewing our article. We have addressed your suggestions/comments as outlined in the attached PDF-file.

Regards

Ralf Weiskirchen 

Reviewer 2 Report

Comments and Suggestions for Authors Dear editors, This article is a comprehensive review that integrates multidisciplinary perspectives and focuses on the "ovarian-liver axis", a new perspective on endocrine metabolism interaction that has received attention in recent years. It provides a systematic review of the association between female liver disease and reproductive endocrine diseases (PCOS, hypogonadotropin, Turner syndrome, menopause). The article integrates multiple dimensions, such as molecular mechanisms, epidemiological evidence, metabolic pathology, immunology, etc., and has strong academic comprehensiveness and practical guidance significance. Introduction Can enhance the historical and definition traceability of the term 'ovary-liver axis'; Suggest clearly stating the "triple objectives of this article" or "scientific problem framework" to enhance guidance. Molecular Mechanisms Suggest adding a summary of the interaction mechanism between estrogen signaling and inflammatory mediators (such as TNF-α, IL-6); It can be emphasized how ovarian hormones reverse the formation of hepatic lipid droplets and insulin sensitivity; In the section on liver metabolism of sex hormones, it is recommended to add "The impact of decreased estrogen clearance ability after menopause". Section 3 Table 1 is information-intensive; it is recommended to use a more concise style (such as a "forest map" to assist in expressing OR range); Additional graphical reinforcement can be provided for the mechanism model of PCOS-induced NAFLD, such as low SHBG and insulin resistance; Emphasizing regional differences, such as East Asia/Middle East, has clinical significance, but it is recommended to separate the discussion of geography/ethnicity into a separate section. Section 4 Suggest adding a mechanistic diagram on whether liver disease in TS patients is caused by estrogen deficiency leading to metabolic disorders; At present, the information in this section is fragmented, and it is suggested to reconstruct it based on the triple dimensions of disease classification, research methods, and discovery; Some of the conclusions in the discussion, such as the role of HRT, are relatively vague. It is suggested to clarify the direction based on actual research data. Section 5 Suggest presenting and strengthening the evidence for the liver protective effects of hormone replacement therapy (HRT) separately; More intuitive illustrations can be used to display the risk changes of NAFLD in women at different menopausal stages; It is suggested to emphasize the causal mechanism between menopausal related fat redistribution and liver metabolic disorders at the end. Section 6 Suggest dividing the three major mechanism chains of "inflammation", "hormone clearance disorders", and "insulin resistance" into sections; Additional discussion can be conducted on the cross mechanism between "novel biomarkers" such as circular disruption and NAFLD/PCOS; Suggest further exploring the potential intervention value of "antioxidant therapy, insulin sensitizers, mTOR inhibitors".

Author Response

Dear Reviewer 2,

Thank you for your valuable comments and the time you spent reviewing our article. We have addressed your suggestions/comments as outlined in the attached PDF-file.

Regards

Ralf Weiskirchen 

Reviewer 3 Report

Comments and Suggestions for Authors

The present manuscript is a review article examining published reports on “The Ovary-Liver Axis.” The authors have focused on relevant recent manuscripts published in English. They examined manuscripts detailing conditions in women with hypogonadism, including genetic, medical, and menopausal bases. Links with liver dysfunction demonstrated ovarian hormonal and reproductive malfunctions, for example,  polycystic ovarian syndrome. The limited number of available studies on the ovarian-liver axis hinders the completeness of the review, which reinforces the conclusion that future research is needed, particularly on causal pathways. Overall, the review is well-written and should be well-received by readers.

Minor criticisms:

L 15 delete aims to and our current understanding

L 19 change to analyzed

L 26 delete there are still

L 27 axis highlight

L 35 delete research shows that

L 46 relevant molecular

L 47 delete observed

L 64-66 NAFLD, NASH, MASLD, MASH, MALFD are not in the list of abbreviations.

L 75 delete identified during our bibliographic research

L 77 delete in the present review

L 80 should a source reference be included (5)?

L 108 delete strategies

L 118 E2 is not in the list of abbreviations

L 129 delete gonadotropins like

L 133 delete reproductive events like

L 148 change levels to concentrations here and elsewhere

L 177 insert a new line to generate Table 3 heading

L 178 subjects exhibited

L 210 replace It is important to note that with Notably

L 233delete it is pinpointed that

L 226 delete it is noted that

L 249 delete its

L 257 conducted: What are consecutive individuals?

Han change frp to from

Viuff beneficial

L 258  Should this list be distinguished from the list at the end?

L 324 insert line to generate Table heading

Veronese  change my to may

Yang When did the study begin?

L 326  Should this list be distinguished from the list at the end?

L 333 insert new paragraph

L 367 change ovar to ovary; increased inflammatory signals

L461 accept change

L473 accept change

L483 accept change

Author Response

Dear Reviewer 3,

Thank you for your valuable comments and the time you spent reviewing our article. We have addressed your suggestions/comments as outlined in the attached PDF-file.

Regards

Ralf Weiskirchen 

Round 2

Reviewer 1 Report

Comments and Suggestions for Authors

The author has addressed my concerns very well.

Reviewer 2 Report

Comments and Suggestions for Authors

The authors have revised a lot accroding to my concerns. I think it is suitable for publication.